Predicting the current fishable habitat distribution of Antarctic toothfish (Dissostichus mawsoni) and its shift in the future under climate change in the Southern Ocean

Liu Jie 1
Zhu Ancheng 1
http://orcid.org/0000-0003-2967-6488 Wang Xitao 1
Zhou Xiangjun 1
Chen Lu 1 2 chenlu6789@stu.ouc.edu.cn
1 Planning and Sea Island Department, Shandong Marine Forecast and Hazard Mitigation Service , Qingdao, Shandong , China
2 Ocean University of China, College of Marine Life Sciences , Qingdao, Shandong , China
Esteban María Ángeles
Electronic publication date: 2024 Mar 29
Publication date: 2024
Volume: 12
Electronic Location ID: e17131
Received 2023 Aug 11; Accepted 2024 Feb 27
Copyright: © 2024 Liu et al.
Copyright year: 2024
Copyright holder: Liu et al.
License: This is an open access article distributed under the terms of the Creative Commons Attribution License, which permits unrestricted use, distribution, reproduction and adaptation in any medium and for any purpose provided that it is properly attributed. For attribution, the original author(s), title, publication source (PeerJ) and either DOI or URL of the article must be cited.
License URL: https://creativecommons.org/licenses/by/4.0/

Keywords: Dissostichus mawsoni, Climate change, Ensemble model, Fishable habitat distribution

Funding: The authors received no funding for this work.

==============================
Global warming continues to exert unprecedented impacts on marine habitats. Species distribution models (SDMs) are proven powerful in predicting habitat distribution for marine demersal species under climate change impacts. The Antarctic toothfish, Dissostichus mawsoni (Norman 1937), an ecologically and commercially significant species, is endemic to the Southern Ocean. Utilizing occurrence records and environmental data, we developed an ensemble model that integrates various modelling techniques. This model characterizes species-environment relationships and predicts current and future fishable habitats of D. mawsoni under four climate change scenarios. Ice thickness, depth and mean water temperature were the top three important factors in affecting the distribution of D. mawsoni. The ensemble prediction suggests an overall expansion of fishable habitats, potentially due to the limited occurrence records from fishery-dependent surveys. Future projections indicate varying degrees of fishable habitat loss in large areas of the Amery Ice Shelf’s eastern and western portions. Suitable fishable habitats, including the spawning grounds in the seamounts around the northern Ross Sea and the coastal waters of the Bellingshausen Sea and Amundsen Sea, were persistent under present and future environmental conditions, highlighting the importance to protect these climate refugia from anthropogenic disturbance. Though data deficiency existed in this study, our predictions can provide valuable information for designing climate-adaptive development and conservation strategies in maintaining the sustainability of this species.

Introduction

Anthropogenic-induced climate changes are continuing to alter the marine ecological environment, such as an increase in water temperature, ocean acidification and a decrease of ice cover in the polar ocean (Perovich & Menge, 2009; Duncan et al., 2022; Falcón & Séférian, 2022). Environmental changes are reshaping marine organisms’ distribution patterns, as these species rely on specific ecological niches (Walther et al., 2009). A number of studies have found the migration of marine species to polar or deeper waters in response to climate change (Fossheim et al., 2015; Spies et al., 2020; Li et al., 2022). Investigating the distribution shifts of marine species is crucial for developing climate-adaptive conservation and management strategies (Abecasis, Afonso & Erzini, 2014; Carman et al., 2016; Paukert et al., 2021).

The Southern Ocean covers roughly 15% of the global ocean surface, but contributes a great deal in absorbing global heats and slowing down the velocity of global warming (Huguenin, Holmes & England, 2022). Climate change has notably altered the Southern Ocean’s marine environment, with water temperatures rising by 0.17 °C from 1950 to 1980, nearly double the global ocean’s average temperature increase (Giorgi, Widmann & Widmann, 2001). A strong ocean current, named Antarctic Circumpolar Current (ACC) flows from east to west and connects the Pacific, Atlantic, and Indian oceans, making the Southern Ocean to be more different from the other oceans. Antarctic marine organisms, having evolved to adapt to this unique environment, may be particularly sensitive to climate-induced environmental changes. For example, Ran et al. (2022) observed a significant reduction in suitable habitats for the Antarctic Jonas fish (Notolepis coatsorum) at its northern range limits due to climate change. Additionally, they predict that this species will expand into new habitats in the Ross Sea, Weddell Sea, and northeastern Antarctic coastal waters. Antarctic krill (Euphausia superba), a crucial species in the Southern Ocean ecosystem, is projected by Lin, Zhao & Feng (2022) to experience a decline in suitable habitats in the Cosmonaut Sea under various socioeconomic scenarios.

The antarctic toothfish, Dissostichus mawsoni (Norman 1937) is a high trophic-level predator and the largest teleost fish endemic to the Southern Ocean, with a circumpolar distribution around the Antarctic continent (Mesa, Riginella & Jones, 2019). As a generalist feeder, the diet of D. mawsoni is diverse and varies with site and ontogenetic stage. Fish are D. mawsoni’s most common prey, although cephalopods and prawns are also occasionally consumed (Hanchet et al., 2015; Yoon et al., 2017; Queirós et al., 2022). Meanwhile, D. mawsoni is the prey of killer whales and Weddell seals (Queirós et al., 2022). D. mawsoni sets up the connection between small fishes at low trophic level and marine mammals at high trophic level, thus occupying an important position in the food web. As an important commercial species, the exploratory fishery of D. mawsoni began in 1997, conducted by the commercial longlines in the Conservation of Antarctic Marine Living Resources (CCAMLR) subarea 88.1 (Hanchet, Mormede & Dunn, 2010). CCAMLR data indicate that by 2018, the annual D. mawsoni catch stabilized at around 2,300 tons, less than Antarctic krill and Dissostichus eleginoides, ranking it as the third largest Southern Ocean fishery (CCAMLR Secretariat, 2023). Considering its critical ecological role and economic value in longline fisheries, assessing the impact of climate change on D. mawsoni’s habitat suitability is essential for the species’ sustainability.

Species distribution models (SDMs) establish species-environment relationships using georeferenced species distribution data (e.g., presence/absence, abundance) and environmental data. It has proved to be an effective method in characterizing the suitable niche of species (Citores et al., 2020), estimating the risk of invasive species (Carman et al., 2016; Poursanidis et al., 2020), and informing conservation strategies (Yoon et al., 2017; Cázares et al., 2021) over the past two decades. The established SDMs based on the current data can forecast the future distribution of species on condition that future environment conditions can be projected based on the global climate models under different greenhouse gas emission scenarios. SDMs are constructed using various statistical algorithms. Integrating these into a multi-algorithm ensemble model overcomes the limitations of individual models, improving predictive accuracy and reducing uncertainty (Robinson et al., 2017). Recently, a growing number of studies have applied the ensemble model to estimate climate impacts on the habitat suitability of demersal deep-sea species, such as Antarctic jonasfish and Pacific cod (Li et al., 2022; Ran et al., 2022). Yates et al. (2019) integrated fishery-dependent and environmental data to investigate the distribution of D. mawsoni along East Antarctica. However, how the climate change to affect the habitat suitability of D. mawsoni under the mid- and long-term effects did not attract enough attention at present.

In this study, we first obtained occurrence records and environment data from online public datasets; we then developed an ensemble model by integrating individual models to characterize species-environment relationships; we finally conducted the ensemble predictions of D. mawsoni fishable distribution under present-day and future environment conditions in the Southern Ocean. This study aims to: (1) identify key factors influencing D. mawsoni’s distribution; (2) assess its fishable distribution shifts by mid and late 21st century under four climate scenarios; (3) identify climate refugia by comparing the fishable habitat distributions of D. mawsoni under current and future environmental conditions. We anticipate our study will inform climate-adaptive conservation and management strategies to sustain D. mawsoni.

Materials and Methods

Presence/pseudo-absence data

The occurrence records of D. mawsoni were downloaded from three public online datasets: the Ocean Biodiversity Information System (OBIS, http://www.iobis.org), the Global Biodiversity Information Facility (GBIF, https://doi.org/10.15468/dl.h33snm), and the Integrated Digitized Biocollection (iDigBio, http://www.idigbio.org). A total of 11,849 occurrence records located in the Southern Ocean between 2000–2014, which corresponded with the time span of environment variables, were obtained. These records were mainly from fishery-dependent sampling and therefore our predictions based on them can only represent the fishable distribution of D. mawsoni. Because sampling was not random, the spatially clustering of occurrence record may over-represent the environment conditions. To mitigate this bias, we used the R package spThin (Lammens et al., 2015) to filter records, retaining one per 5 × 5 arc min grid (approximately 9.2 × 9.2 km at the equator), aligning with the environmental data’s spatial resolution. After this data thinning process, 451 occurrence records were retained for the model analysis (Fig. 1).

Figure 1 The predicted habitat suitability of Dissostichus mawsoni under current environment conditions.

The available occurrence records were showed by pink points in the map. The colors, from blue to red, indicated the habitat suitability from low to high. The five lines indicated the five Antarctic Circumpolar Current fronts. NB, northern boundary; SAF, Subantarctic Front; PF, Polar Front; SACCF, Southern Antarctic Circumpolar Current Front; SB, southern boundary.

Following Iturbide, Bedia & Gutiérrez (2018), we used a three-step method to generate pseudo-absence records, tenfold the number of occurrence records. The initial step involved utilizing a presence-only support vector machine algorithm to define environmentally unsuitable regions and then limited pseudo-absence records to these unsuitable regions. The subsequent step consisted of constructing SDMs using pseudo-absences within buffers of varying extents encircling the occurrence sites. The third step focused on identifying the optimal buffer distance informed by the performance of the SDMs established in the preceding step. This method balances the use of spatial and environmental dimensions for the selection of pseudo-absence points and has been proven to be superior to other methods.

Environmental predictor variables

Selecting appropriate predictor variables is crucial for the effectiveness of Species Distribution Models (SDMs). Considering data availability and species-environment relationships, ten environmental predictors were initially selected. Mesa, Riginella & Jones (2019) found that juvenile D. mawsoni was not uniformly distributed along the East Antarctica and its spatial variations involved complex interactions with bathymetry, temperature, sea ice and salinity. These variables were thus chosen as predictors. As a high-trophic level predator, the spatial distribution of D. mawsoni is greatly affected by the variations in the abundance and distribution of prey. D. mawsoni is not herbivorous so chlorophyll concentration cannot affect its distribution directly, but this factor may affect the prey availability through the bottom-up effects. Therefore, this factor was involved in the model analysis. D. mawsoni, predominantly found on continental shelves and slopes, has a circumpolar distribution. Mature adults prefer deeper waters for spawning (Hanchet et al., 2015), making seabed slope, ruggedness, and Bathymetric Position Index (BPI) relevant to their distribution. These factors were calculated from the depth layer utilizing ArcGIS Pro v3.0 and then were incorporated into our model analysis. The slope grid was generated using the integrated Spatial Analyst tool. Ruggedness and BPI were calculated using the Benthic Terrain Model 3.0 tool. For ruggedness, a neighborhood size of five grid cells was applied, while BPI was calculated with an inner radius of one and an outer radius of five grid cells, in accordance with the methodology outlined by Walbridge et al. (2018). Ocean circulation primarily affects the early life-history stages by transporting eggs, larvae, and juvenile Antarctic toothfish from the offshore spawning areas to the coastal recruitment areas (Behrens et al., 2021). It also facilitates adults to swim along the shelf slope and back into the spawning grounds (Ashford et al., 2012). Therefore, we included current velocity as a predictor.

Multicollinearity among predictors can lead to model over-parameterization, compromising predictive accuracy (Ran et al., 2022). To address this, we conducted a variance inflation factor (VIF) analysis. The predictor manifesting a value of VIF over 10 was systematically excluded (O’Brien, 2007). After this process, five terrain variables (depth, distance to shore, slope, ruggedness and BPI) and five temporally dynamic environmental variables (current velocity, salinity, mean temperature, mean chlorophyll concentration and ice thickness) were retained for the model analysis (Table 1). Additionally, these variables exhibited low paired Pearson correlation coefficients (Fig. 2).

Table 1 Summary statistics of current environment conditions used to model the Dissostichus mawsoni distribution.

Variable	Min	Max	Mean	Unit	Source	
Mean current velocity	0.0008	0.38	0.06	m/s	Bio-ORACLE	
Mean salinity	32.53	35.54	34.67	PSS	Bio-ORACLE	
Ice thickness	0.00	1.61	0.18	m	Bio-ORACLE	
Mean water temperature	−1.84	15.77	0.47	°C	Bio-ORACLE	
Mean chlorophyll concentration	0.004	2.25	0.02	mg/m3	Bio-ORACLE	
Water depth	−7,873.42	0	−3,699.72	m	MARSPEC	
Distance to shore	1.00	2,774.00	901.93	km	MARSPEC	
Fine-scale benthic positioning index (BPI)	−2,379.00	3,034.00	−0.45	/	Calculated from depth	
Ruggedness	−0.0000019	0.0090	0.0001	/	Calculated from depth	
Slope	0.0009	11.47	0.60	degree	Calculated from depth	

Figure 2 Pearson correlation coefficients of 10 environment variables retained for the model analysis.

Dep, depth; BPI, fine-scale Bathymetric Position Index; Rug, seafoor ruggedness; Slope, Seafloor Slope; DTS, distance to land; Chlmean, mean chlorophyll concentration; CV, current velocity; Sal, salinity; Tmean, mean water temperature; IT, ice thickness.

The Bio-ORACLE v2.2 dataset (http://www.bio-oracle.org) (Assis et al., 2018) offers environmental data for current (2000–2014 averages) and future periods (2050s: 2040–2045 averages, 2100s: 2090–2100 averages) under various greenhouse gas emission scenarios, with a 5 × 5 arc min spatial resolution. The future environment data were projected based on three atmospheric-ocean general circulation models (AOGCMs: CCSM4, HadGEM2-ES and MIROC5). In order to reduce model-dependent uncertainty, we used the average values of predictions from the three AOGCMs to represent the future environment conditions. We considered four Representative Concentration Pathways (RCPs): RCP 2.6, RCP 4.5, RCP 6.0 and RCP8.5, which indicate the continually increased concentrations of greenhouse gas emissions. Depth and distance to shore, sourced from the Global Marine Environment Datasets (Basher, Bowden & Costello, 2018), were assumed to remain constant over time.

Modelling process

Utilizing presence/pseudo-absence records and current environment data, we developed SDMs to relate the occurrence probability of species with environment conditions using the R package biomod2 (Thuiller et al., 2014). The package incorporates ten widely-utilized modeling algorithms: Surface Range Envelope (SRE), generalized linear model (GLM), generalized additive model (GAM), multiple adaptive regression splines (MARS), generalized boosting model (GBM), flexible discriminant analysis (FDA), classification tree analysis (CTA), Random Forest (RF), artificial neural network (ANN), and maximum entropy (Maxent). The predictive ability of each model was evaluated by a five-fold cross validation method in which 80% of data were selected randomly to train the model and the rest 20% of data used to validate the model (Guisan, Thuiller & Zimmermann, 2017). This procedure was iterated 10 times. Area under the curve (AUC) and true skill statistics (TSS) were used to assess model performance. To reduce model-dependent uncertainty, we used the AUC value of each individual model as weight to build an ensemble model. Models exhibiting TSS > 0.8 and AUC > 0.9 were chosen to build the ensemble model (Araújo et al., 2005).

Predictor importance and response curves

The importance of each predictor was evaluated using the random isolation method included in the biomod2 package. This method involved calculating the Pearson correlation coefficient between predictions made using all predictors and those using a randomly permuted evaluated predictor (Guisan, Thuiller & Zimmermann, 2017). High correlation signified low importance of the evaluated predictor. Response curves were plotted to visualize how occurrence probability varied with each environmental predictor’s gradient.

Predictions of current and future fishable D. mawsoni distribution

The ensemble model developed based on the current environment data was used to predict the current fishable D. mawsoni distribution and future fishable distribution under four (RCP 2.6, RCP 4.5, RCP 6.0 and RCP8.5) climate change scenarios in the 2050s and 2100s. The prediction from the model at each raster grid indicates the habitat suitability with a value between 0–1. To facilitate interpretation, continuous habitat suitability values were converted to binary (0/1) values. The grids with the habitat suitability greater than the cut-off value which maximizes the probability threshold of TSS were converted to 1 and the others were converted to 0 (Guisan, Thuiller & Zimmermann, 2017). Using the current fishable habitat distribution as a reference, the future habitat change in each raster grid has three situations: present now and in the future (stable), present now but predicted to be lost in the future (loss), and absent now but predicted to be present in the future (gain).

Results

Model performance

TSS values ranged from 0.51 (SRE) to 0.87 (GBM), while AUC values varied from 0.75 (SRE) to 0.98 (RF) across the ten models. Both TSS and AUC revealed significant variations in predictive ability among the models (Fig. 3). Five models-GBM, GAM, CTA, RF, and MAXNET-were chosen for the ensemble model based on their TSS and AUC cut-off values. The TSS and AUC values of the ensemble model were 0.922 and 0.994 respectively, greater than any of the individual models, indicating an improvement of model performance.

Figure 3 The ability of 10 individual models in predicting the habitat suitability of Dissostichus mawsoni, estimated by AUC and TSS values.

TSS, True Skill Statistics; AUC, Area Under the receiver operating characteristic Curve. Data are expressed as mean ± standard error. Dashed lines indicated the cutoff values of TSS and AUC used to build ensemble model.

Predictor importance and response curves

Ten environment predictors were included in the ensemble model to predict the potential fishable D. mawsoni distribution. Based on a randomization method, ice thickness was proved to be the most influential factor to the D. mawsoni distribution, followed by depth. Mean temperature, distance to shore, salinity and mean chlorophyll concentration showed a similar, moderate importance and the remained factors including slope, ruggedness, BPI and current velocity presented little influence on the D. mawsoni fishable distribution (Fig. 4). Response curves depicting the relationship between occurrence probability and environmental gradients are presented in Fig. 5. Occurrence probability sharply increased with ice thickness, peaking at 0.25–0.75 m, before gradually declining. Occurrence probability increased as depth decreased, stabilizing at approximately 0.73 in depths of 1,000–2,000 m. Occurrence probability surged with rising water temperature, peaking at 1.5 °C, and then sharply declined, bottoming out at temperatures above 5 °C. Salinity, mean chlorophyll concentration, current velocity, distance to shore, ruggedness, and BPI each differently affected the fishable distribution, whereas slope changes had no discernible impact on D. mawsoni’s occurrence probability.

Figure 4 The relative importance of 10 predictor variables in predicting the habitat distribution of Dissostichus mawsoni based on the ensemble model.

Dep, depth; BPI, fine-scale Bathymetric Position Index; Rug, seafoor ruggedness; Slope, Seafloor Slope; DTS, distance to land; Chlmean, mean chlorophyll concentration; CV, current velocity; Sal, salinity; Tmean, mean water temperature; IT, ice thickness.

Figure 5 Response curves of the occurrence probability of Dissostichus mawsoni along the environment gradients.

Present-day fishable habitat distribution

The prediction from the ensemble model under current environment condition (Table 1) showed that the fishable habitats of D. mawsoni concentrated around the peripheral seas of the Antarctic continental shelf (Fig. 1). Substantial suitable fishable habitats were located in the Amundsen Sea, Ross Sea, Amery Ice Shelf, and Bellingshausen Seas. Areas predicted to have high habitat suitability encompassed most occurrence points. Relatively low habitat suitability was observed in the coastal waters of Northern Antarctica, particularly in the Weddell Sea.

Future fishable habitat distribution and its shift under future environment conditions

Environmental conditions for the 2050s and 2100s under four emission scenarios are presented in Table 2. Under all emission scenarios, current velocity and salinity exhibited minimal changes in both the 2050s and 2100s. As expected, mean water temperature showed a substantial rise and ice thickness showed a substantial decrease in the future. The extent of these changes intensified with higher emission concentrations. Unexpectedly, chlorophyll concentration increased in future projections across all emission scenarios.

Table 2 Current environment conditions and the average values of climatic change in 2050s and 2100s under four Representative Concentration Pathways (RCPs) scenarios in the Southern Ocean.

Variable	Current	2050	2100	
RCP 2.6	RCP 4.5	RCP 6.0	RCP 8.5	RCP 2.6	RCP 4.5	RCP 6.0	RCP 8.5	
Mean current velocity	0.06	0	0	−0.01	−0.02	−0.01	−0.01	−0.01	−0.04	
Mean salinity	34.67	0	+0.01	+0.01	+0.01	+0.01	+0.01	+0.01	+0.01	
Ice thickness	0.18	−0.01	−0.02	−0.02	−0.04	−0.03	−0.04	−0.05	−0.09	
Mean water temperature	0.47	+0.12	+0.13	+0.18	+0.20	+0.25	+0.29	+0.32	+0.37	
Mean chlorophyll concentration	0.02	+0.20	+0.23	+0.20	+0.19	+0.23	+0.19	+0.17	+0.22	
Water depth	−3,699.72	/	/	/	/	/	/	/	/	
Distance to shore	901.93	/	/	/	/	/	/	/	/	
Fine-scale benthic positioning index (BPI)	−0.45	/	/	/	/	/	/	/	/	
Ruggedness	0.0001	/	/	/	/	/	/	/	/	
Slope	0.60	/	/	/	/	/	/	/	/	

The fishable habitat suitability in the 2050s and 2100s under the four emission scenarios were shown in Figs. 6 and 7. The areas with different habitat suitability were distributed in the nearshore waters encircling the Antarctic continent. In the 2050s, substantial suitable habitats were located near the Antarctic Peninsula, in the Bellingshausen Sea, the Ross Sea, and around the Amery Ice Shelf. The extent of highly suitable areas expanded in 2010, with the degree of expansion varying according to the emission scenarios.

Figure 6 Habitat suitability maps predicted by the ensemble model in the 2050s (2040–2050) under four Representative Concentration Pathways (RCPs): RCP 2.6, RCP 4.5, RCP 6.0 and RCP8.5 scenarios.

Figure 7 Habitat suitability maps predicted by the ensemble model in the 2100s (2090–2100) under four Representative Concentration Pathways (RCPs): RCP 2.6, RCP 4.5, RCP 6.0 and RCP8.5 scenarios.

Compared with the current distribution of D. mawsoni, the ensemble model predicted a medium-term (2050s) expansion of suitable habitat ranging from +9.35% (RCP 4.5) to +12.38% (RCP 8.5), and a long-term (2100s) expansion from +13.48% (RCP 2.6) to +19.81% (RCP 6.0) (Table 3). Predictions indicate that D. mawsoni’s fishable habitats will extend to new areas near the Weddell Sea and the Ross Sea (Figs. 8 and 9). The extent of this expansion varies with time and greenhouse gas emission scenarios. It is noteworthy that the range of suitable fishable habitat around the Amery Ice Shelf will decrease gradually in the future (Figs. 8 and 9). Under all emission scenarios, suitable habitats in the Ross Sea are expected to persist through the 2050s and 2100s.

Table 3 Change (%) of fishable distribution range of Dissostichus mawsoni under four Representative Concentration Pathways (RCPs) scenarios in 2050s and 2100s.

RCP conditions	2050s (2040–2050)	2100s (2090–2100)	
2.6	+9.35	+13.48	
4.5	+10.03	+12.20	
6.0	+12.38	+19.81	
8.5	+10.25	+9.43	

Figure 8 Fishable habitat changes predicted by the ensemble model in the 2050s (2040–2050) under four Representative Concentration Pathways (RCPs): RCP 2.6, RCP 4.5, RCP 6.0 and RCP 8.5 scenarios.

Figure 9 Fishable habitat changes predicted by the ensemble model in the 2100s (2090–2100) under four Representative Concentration Pathways (RCPs): RCP 2.6, RCP 4.5, RCP 6.0 and RCP 8.5 scenarios.

Discussion

Ongoing global warming continues to affect marine communities and alter ecosystem functions (IPCC, 2022; Trathan & Agnew, 2010; Wassmann et al., 2011). Fish endemic to the Southern Ocean are potentially more vulnerable to global climate change due to their narrow ecological niches and limited colonization space (Freer et al., 2019). Consequently, we aimed to assess the impact of climate change on D. mawsoni’s habitat suitability by integrating species occurrence records with environmental data, to guide sustainable development and management strategies.

Model performance and limitations

The ensemble model consistently outperformed individual models, as indicated by TSS and AUC values, suggesting that integrating individual models to create a weighted average enhances predictive capabilities. Previous studies corroborate the superiority of ensemble models over individual models in evaluating climate change impacts on marine demersal species (Behrens et al., 2021; Li et al., 2022; Ran et al., 2022). In this study, the SRE model exhibited the poorest performance, likely because it solely relies on presence data, unlike other models which utilize both presence and absence data. This underscores the need for simulating pseudo-absence records to enhance model performance in the absence of true absence data.

Despite the aforementioned advantages of the ensemble model, there were limitations in the data used for its development. First, the occurrence records used in this study, sourced from online public datasets, predominantly relied on fishery-dependent observations. Therefore, our study’s predictions primarily represent the fishable distribution of D. mawsoni. Additionally, the fishery-dependent data held by CCAMLR, owned by multiple countries, necessitates obtaining permission from all data owners for specific analyses. It is hard for us to gain permission for using CCAMLR data to conduct our analysis. The limited number of occurrence records may inadequately represent D. mawsoni’s realized niche, thus impacting the SDM’s predictive accuracy. Second, treating simulated pseudo-absence records as true absences poses certain risks. Pseudo-absences might inaccurately represent true absences, potentially due to insufficient sampling at certain sites. When true absence data are available, the SDM predictions based on presence/absence data are more accurate than those based on presence only or presence/pseudo-absence data (Xu et al., 2022). Third, given that adult D. mawsoni predominantly inhabit the sea bottom, sediment characteristics like median grain size and type may influence their distribution. Meanwhile, as a commercial-target species, human fishing activities could also affect the distribution of D. mawsoni. Failing to include substrate and anthropogenic variables due to data unavailability in the SDM will minimize the model predictive ability. Fourth, inter-annual environmental variations, such as the timing of ice emergence and retreat, influence D. mawsoni recruitment (Behrens et al., 2021). However, due to data constraints, we used multi-year (2000–2014) average environmental data to represent current conditions, thereby obscuring inter-annual variations and potentially reducing SDM predictive accuracy. Fifth, it is essential to recognize that dispersal ability significantly affects D. mawsoni’s distribution. Lenoir et al. (2020) discovered that marine species adapt to suitable niches in response to climate warming six times faster than terrestrial species, attributed to the less constrained physical environment for dispersal in marine habitats. D. mawsoni is not a powerful swimmer and actually, most individuals move few kilometers throughout their entire lives (CCAMLR Secretariat, 2023). Variable ocean currents and complex bottom topography may impede this species from reaching all suitable habitats. Consequently, our study might overestimate D. mawsoni’s potential future fishable distribution under the assumption of unlimited dispersal.

Response of D. mawsoni to environment change

The species-environment relationships characterized in this study can provide key information to understand the response of D. mawsoni to environment change. Ice thickness is the most important factor in shaping the distribution of D. mawsoni. Parker et al. (2021) indicated the significant role of sea ice in D. mawsoni’s early life history. Consequently, global warming-induced changes in ice thickness will substantially impact this species’ population size and distribution. Our findings reveal that D. mawsoni’s occurrence probability sharply increases with ice thickness values below 0.25 m and peaks at 0.25–0.75 m, suggesting that specific sea ice thickness levels are vital for the species. While the upper limit of optimal ice thickness for D. mawsoni remains unconfirmed, the reduced occurrence probability above 0.75 m could be attributed to sampling bias, as thicker ice impedes fishing boats’ access, resulting in fewer captures. Yates et al. (2019) found a similar shape of response curve with ours in studying the relationship between catch rate of D. mawsoni and mean ice cover based on fishery-dependent and environment data from years 2003–2017 along East Antarctica. Ice thickness is most likely to positively correlate with mean ice cover and the similar findings confirmed the importance of sea ice for D. mawsoni.

This study proved that water depth was the second important factor in shaping the distribution patterns of D. mawsoni. The previous studies in the Ross Sea and the eastern Antarctica demonstrated that the mean weight and proportion of adults increased with the increasing depth, implying a gradual migration from shallow to deep waters as fish grows (Mesa, Riginella & Jones, 2019). The Ross Sea contained important spawning grounds and the older and larger fishes spawn on the ridges, banks and seamounts of the Pacific-Antarctic Ridge to the north and east of the Ross Sea with deeper waters (Parker et al., 2019). The southern part of the Amundsen Sea was considered to be an important source of immature fishes based on the analysis of fishery data in this area (Abecasis, Afonso & Erzini, 2014; Mekonnen & Hoekstra, 2014). The acquired occurrence records were mainly located in the west of Antarctic Peninsula, the Amundsen Sea and the Ross Sea, resulting in a larger proportion of juvenile records. The preference of younger cohorts for shallower waters likely explains the observed increase in occurrence probability with decreasing depth in this study. Future work should focus on characterizing species-environment relationships, taking into account sex and ontogenetic stages.

Habitat change in the future and implication for management

Beyond our expectation, a fishable habitat expansion was predicted under all emission scenarios in the future. We speculate this result may stem from the limited scope of available occurrence records, primarily derived from fishery-dependent surveys, covering only a fraction of D. mawsoni’s distribution. The occurrence records were primarily located in the coast of West Antarctica, while fishing activities were also conducted in the East Antarctica in addition to in the Ross and Amundsen Sea of the West Antarctica. Due to the harsh weather conditions in the Southern Ocean, the extensive fishery-independent surveys were restricted by financial and logistical terms. Additionally, while regions with thicker ice may be suitable for D. mawsoni, such ice impedes fishing boats’ access. The lack of sampling in the habitats with thicker ice is probably the reason for the predictive results of future habitat expansion. Although the data deficiency, our predictions can provide valuable knowledge for informing conservation measures and management strategies in maintaining the sustainability of D. mawsoni. The climate of the Antarctic Peninsula is the most rapidly changing in the Southern Hemisphere and marine species in this region are extremely sensitive to their environment, with predictive species and population removal in the face of a little rise in the water temperature (Mao et al., 2021). Predictions for West Antarctica were more robust due to the greater data availability in this region. Large amounts of suitable habitats in the West Antarctic Peninsula were predicted to be lost in the face of future climate change, which was consistent with a similar study on the Antarctic jonasfish (Ran et al., 2022). Parker et al. (2019) speculated that seamounts in the northern Ross Sea are the spawning grounds of D. mawsoni. The fishable habitats in the Ross Sea, including the spawning grounds in the deeper waters, were predicted to persistent under future environment conditions, highlighting the importance to protect these climate refugia from anthropogenic disturbance.

Supplemental Information

Supplemental Information 1 Presence/pseudo-absence data.

Supplemental Information 2 R scripts for VIF analysis and pseudo-absence records generation.

Supplemental Information 3 R scripts for building SDM.

We greatly appreciate Global Biodiversity Information Facility (GBIF, https://www.gbif.org), Ocean Biogeographic Information System (OBIS, https://obis.org), and the Integrated Digitized Biocollection (iDigBio, http://www.idigbio.org) for providing online data to conduct this study.

Additional Information and Declarations

Competing Interests

Author Contributions

Data Availability

The authors declare that they have no competing interests.

Jie Liu conceived and designed the experiments, performed the experiments, analyzed the data, prepared figures and/or tables, authored or reviewed drafts of the article, and approved the final draft.

Ancheng Zhu performed the experiments, prepared figures and/or tables, and approved the final draft.

Xitao Wang performed the experiments, analyzed the data, prepared figures and/or tables, and approved the final draft.

Xiangjun Zhou analyzed the data, authored or reviewed drafts of the article, and approved the final draft.

Lu Chen conceived and designed the experiments, authored or reviewed drafts of the article, and approved the final draft.

The following information was supplied regarding data availability:

The code and raw absence/pseudo-absence data are available in the Supplemental Files.

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
