# Peer review of "Predicting the current fishable habitat distribution of Antarctic toothfish (Dissostichus mawsoni) and its shift in the future under climate change in the Southern Ocean"

_PeerJ, doi:10.7717/peerj.17131_

## Round 0.1 · original submission · Major Revisions

The work is interesting although it requires a thorough revision in order to improve it. There are many grammatical errors throughout the manuscript, hence the whole text should be revised and corrected. In addition, the conclusions presented are not based on the results obtained, therefore, they should be redone in a way that is more faithful to those results. All suggestions made by the reviewers will have to be taken into account in the modified version of the manuscript.

**Language Note:** The Academic Editor has identified that the English language must be improved. PeerJ can provide language editing services - please contact us at copyediting@peerj.com for pricing (be sure to provide your manuscript number and title). Alternatively, you should make your own arrangements to improve the language quality and provide details in your response letter. – PeerJ Staff

·

Basic reporting

The use of english is of adequate quality and scientific level

The authors have provide sufficient background and context while the cited literature references provide update informatiob.

The figures require to be larger since figures 6 and 7 are hardly readable. Suggest to split into two picture per line and provide picture by picture as high resolution figure and geotiff product,

The raw data have not been shared but the DOI from GBIF dataset is provided and is still valid (https://www.gbif.org/occurrence/download/0383709-210914110416597, access at 28/8/2023).

Experimental design

In the analysis, i suggest the use of VIF analysis instead of the Pearson as is more powerfull and examines the relations of the predictors not one by one but one against all in the spatial domain. Also, VIF indicates what percentage of the variance is inflated for each coefficient.

Bio-oracle release the version 2.2. https://www.bio-oracle.org/release-notes-2-2.php and is suggested to use that along with the new SSPs instead of RCPs which are discontinuued by the IPCC.

Also, provide the full code with the data for reproduction of the analysis. This has to be open also if the article will be accepted for publication for the shake of open science.

Validity of the findings

Since the authors use an old version of the environmental predictors, there is a need to repeat the analysis with the updated version. By that, the findings have to be re-evaluated afterwards.

Reviewer 2 ·

Basic reporting

1. There are many grammatical errors throughout the manuscript, in every section and sub-section. Most of the errors are minor, in that the meaning of the sentence is clear, just not grammatically correct. However, these minor errors may cause a barrier to understanding for non-native English speakers.
2. There are a number of sentences for which the meaning is not clear either because of grammatical errors or poor choice of words.
a. LINE 281 – 282
b. LINE 315 – 317
c. LINE 221 – 229 (This single sentence is too long)
3. “data superiority” is an ambiguous term and used on multiple occasions. What makes the data superior? Quantity, resolution, reliability, accuracy…
4. There are a number of field and/or region specific words used that are not standard and need correcting.
a. E.g. “South Ocean” should be Southern Ocean
b. Check the use of “Antarctica” vs “Antarctic”
5. Multiple instances where references or additional references are required to support statements:
a. LINE 36
b. LINE 65
c. LINE 72 (reference relevant CCAMLR report)
d. LINE 255
e. LINE 305
f. LINE 338
6. Incorrectly including in-text references without the year of publication in parenthesis:
a. LINE 52
b. LINE 90
c. LINE 133
d. LINE 320
7. Inclusion of in-text references with no relevance to the statement with which they are associated:
a. LINE 92
8. Repeated inclusion of a non-primary literature reference with no relevance to this topic “B, A., 2008. Heading to the Herbarium. Nature Geoscience. 1, E15-E15. 370 http://doi.org/10.1038/ngeo301.”
a. LINE 292
b. LINE 302
9. The article is well structured in a clear and logical order that tells a coherent story.
10. The figures are all relevant and clear.
11. The tables are clear summaries of various aspects of the data and/or results
12. There is a strange artifact in the .png file for figure 5 at depth values below 0.
13. The manuscript is a self-contained piece of research which clearly outlines their aims in the introduction and tackles them one-by-one in a logical and consistent order within the methodology, results, and discussion section.

Experimental design

1. The topic of this study is timely and gives a good explanation of the knowledge gap it aims to fill, although it’s uniqueness (LINE 258) is overstated (see Cheung et al., 2008).

2. The SDM methods employed were of a high technical standard, using a rigorously tested ensemble approach and testing for parameter co-linearity.

3. I have serious concerns as to how the authors have biased the biological data upon which all their findings are based. In LINE 124 – 128 the authors state their inclusion of pseudo-absences only outside the distribution maps of Hanchett et al. (2015). As Hanchett et al. (2015) mapped this distribution based on fishery observations, attributing pseudo absences only to areas beyond fishery activity emphasizes the biasing effect that having fishery-dependent only presence observations already has. I would like to see the authors support this geographically constrained application of pseudo absences with robust literature references and/or a sensitivity analyses of the results to the placement of pseudo absences.

4. Although it would be possible to replicate the methods of this study, it would be made much easier if the r scripts used were made publicly available in GitHub or other repositories. Likewise, the data used is publicly available, but replication would be easier if their subset biological data and psudo-absence data were made similarly available online.

Validity of the findings

1. The validity of this study is questionable because the main conclusions and their ability to carry out their aims are severely inhibited by the data upon which their SDMs are based.

2. They claim to assess the distribution and suitable habitat of D. mawsoni, but because they are limited to fishery-dependent observations, they cannot claim to assess the full extent of D. mawsoni distribution or predict changes to their habitat. They acknowledge the temporal, spatial, and socio-economic limitations on fishing effort but still claim to predict an expansion of D. mawsoni habitat. Their results are more representative of changes to 'fishable D.mawsoni habitat'. The authors acknowledge that the habitat expansion results were “probably caused by the limited available occurrence records, which could only encompass part of the distributed areas of D. mawsoni.”. I therefore recommend re-wording all mention of D.mawsoni ‘habitat’ or ‘distribution’ to ‘fishable habitat’ or ‘fishable distribution’ respectively.

3. Another major concern is that the authors did not include all the relevant available data to inform their SDMs. There is a large amount of additional fishery dependent observations on D.mawsoni available upon request from CCAMLR. It is unclear how this data would affect their results considering the methods used to mitigate spatial bias in sampling but its omission is unexplained.

4. The authors acknowledge the importance of topographic environmental variables and claim the data is lacking (LINE 143 – 144). However, this data is by no means unavailable since they have depth data. There are a number of readily available packages within r or GIS that can calculate topographic features such as slope, rugosity, etc. from the depth data they have. There is no reason to not calculate and include topographic variables in their analyses.

5. A principal conclusion of this study is that there will be a general expansion of D.mawsoni habitat under future climate scenarios. This conclusion is strongly influenced by the thickness of sea ice (LINE 314), which will decrease under these future scenarios and, they claim, make larger areas of the Antarctic shelf more suitable habitats. They provide no evidence or explanation for thick sea-ice making an area unsuitable for D.mawsoni, only that it makes it inaccessible to fishing (LINE 319-320). It is likely that sea-ice is an important component of D.mawsoni life-cycle (see Parker et al., 2021) and would be expected to have a positive relationship with habitat suitability if the data were unbiased.

Additional comments

Although the scope and SDM methods of this study are sound, I believe that the data upon which the results are based, are not sufficient for the conclusions the authors have made.

There are often limitations in terms of species observations in this region, but the authors have further limited their data by not incorporating CCAMLR observations. Additional environmental (e.g. topographic) data should also be included to overcome the limitations that they recognise within their discussion. Finally, the limitations of using fishery-dependent observations should be reflected throughout the manuscript by recognising that the results do not represent the true habitat or species distributions, but instead represent the 'fishable habitat or species distribution'.

I would recommend a complete re-analysis using the same ensemble SDM methods but with additional environmental and species observation data, as well as a more careful consideration of biases from fishery-dependent data explicitly incorporated into the variable selection process and presentation of the results.

---

## Round 0.2 · accepted · Accept

Many thanks for addressing all the reviewers' suggestions in the revised version of your manuscript. I am pleased to confirm that your paper has been accepted for publication in PeerJ.

Thank you for submitting your work to this journal.

·

Basic reporting

The authors have respond to all previous comments and no issues have remain unsolved.

Experimental design

The authors have followed all suggestions on the analysis of the data. No issues have remain unsolved.

Validity of the findings

Since the authors have used the updated version of the predictor's database and tansparency on the analytical workflow through the access to the code and data, no other issues exist.

Additional comments

N/A

Reviewer 2 ·

Basic reporting

Since my previous review of this manuscript the grammar has been greatly improved and all previous outstanding issues i had have been resolved

Experimental design

As I stated previously "The SDM methods employed were of a high technical standard". My concerns about the biasing affects of the pseudo-absence data used have been addressed by justifying and testing other approaches to generating pseudo-absence data. The inclusion of the scripts used to carry out the analyses is also very welcome.

Validity of the findings

By adjusting the conclusions of this manuscript to acknowledge that they are pertaining to the 'fishable habitat' / 'fishable distribution' they have addressed my principal concern.

I am sorry that the authors were unable to incorporate CCAMLR data. There acknowledgement of this in the manuscript discussion is welcome and justifies their use of a more limited dataset.

There inclusion of topographic environmental parameters made the study more comprehensive, even if it did little to change their results.

Additional comments

I am thoroughly impressed by how comprehensively the authors have addressed all of the reviewer comments.